# From Shunt to Recovery: A Multidisciplinary Approach to Hydrocephalus Treatment in Severe Acquired Brain Injury Rehabilitation

**DOI:** 10.3390/brainsci12010003

**Published:** 2021-12-21

**Authors:** Giovanna B. Castellani, Giovanni Miccoli, Francesca C. Cava, Pamela Salucci, Valentina Colombo, Elisa Maietti, Giorgio Palandri

**Affiliations:** 1Montecatone Rehabilitation Institute, Imola, 40026 Bologna, Italy; francesca.cava@montecatone.com (F.C.C.); pamela.salucci@montecatone.com (P.S.); valentina.colombo@montecatone.com (V.C.); 2Department of Neuroscience, Reproductive and Odontostomatological Sciences, Division of Neurosurgery, Università degli Studi di Napoli “Federico II”, 80138 Naples, Italy; gmiccoli9013@gmail.com; 3Department of Biomedical and Neuromotor Sciences, Alma Mater Studiorum University of Bologna, 40126 Bologna, Italy; elisa.maietti2@unibo.it; 4Department of Neurosurgery, Institute of Neurological Sciences of Bologna IRCCS, Bellaria Hospital, 40139 Bologna, Italy; giopalandri@gmail.com

**Keywords:** rehabilitation outcome, brain injuries, hydrocephalus, ventriculoperitoneal shunt, cerebrospinal fluid

## Abstract

Background: Hydrocephalus among Severe Acquired Brain Injury (SABI) patients remains overlooked during rehabilitation. Methods: A retrospective cohort study was carried out of traumatic and non-traumatic SABI patients with hydrocephalus, consecutively admitted over 9 years in a tertiary referral specialized rehabilitation hospital. Patients were treated with ventriculoperitoneal shunt before or during inpatient rehabilitation and assessed using the Level of Cognitive Functioning Scale and Disability Rating Scale. Logistic regression models were used to identify predictors of post-surgical complications. Linear regression models were used to investigate predictors of hospital length of stay (LOS), disability, and cognitive function. Results: Of the 82 patients, 15 had post-surgical complications and 16 underwent cranioplasty. Shunt placement complication risk was higher when fixed vs. when programmable pressure valves were used. A total of 56.3% achieved functional improvement at discharge and 88.7% improved in cognitive function; of the 82 patients, 56% were discharged home. In multiple regression analyses, higher disability at discharge was related to cranioplasty and longer LOS, while poorer cognitive function was associated with cranioplasty. Increase in LOS was associated with increasing time to shunt and decreasing age. Conclusions: A significant improvement in cognitive and functional outcomes can be achieved. Cranioplasty increased LOS, and fixed pressure valves were related to poorer outcomes.

## 1. Introduction

Severe Acquired Brain Injury (SABI) includes a variety of traumatic or non-traumatic acute brain lesions characterized by the onset of a variably long-lasting state of coma (Glasgow Coma Scale—GCS ≤ 8) together with simultaneous motor, sensory, cognitive and/or behavioral impairment. Non-traumatic SABI arises from hemorrhages (e.g., intracerebral, intraventricular, subarachnoid), infections, brain tumors, anoxia and toxic-metabolic encephalopathy [1,2]. The special goal of inpatient rehabilitation in patients with seriously impaired consciousness is to allow functional recovery that minimizes the functional impact on residual impairments [1,2,3]. Once transferred to inpatient rehabilitation, the management of these patients is often severely uphill. Medical complications may hinder therapeutic efforts, and patients are usually unable to report their symptoms [3,4,5].

Hydrocephalus rate during inpatient rehabilitation ranges from 30% to 86% (between 3 and 12 months after severe brain injury [3,6]). The late occurrence of hydrocephalus and its implications on overall disability are important but poorly studied topics.

The diagnosis of hydrocephalus in SABI patients is in fact frequently challenging. The main criteria include a persistent disorder of consciousness, worsening or plateaued clinical recovery discordant with injury severity, elevated cerebrospinal fluid (CSF) opening pressure, or clinical improvement in behavioral or motor capacities after (high-volume) lumbar puncture [4,5].

Hydrocephalus could reproduce signs and symptoms of acute hydrocephalus with either rising in intracranial pressure, headache, papilledema, nausea and vomit and a progressively rapid decline in the state of consciousness. This happens in CSF pathway blockage (i.e., for intracerebral hemorrhage, intraventricular hemorrhage, subarachnoid hemorrhage, mass effect) [7,8].

At the same time, patients may present with symptoms mimicking Normal Pressure Hydrocephalus (NPH) such as cognitive impairment, gait disturbances and urinary incontinence. In these cases, alterations of CSF flow dynamics as in CSF reabsorption at arachnoid granulations and physiological changes in intracranial venous outflow and pulsatility and/or impairment of the glymphatic pathway are supposed to come into play [7,8,9,10,11].

Post-traumatic hydrocephalus (PTH) is diagnosed in about 0.7 to 29% (but it has been reported to be as high as 51%) of severe traumatic brain injury patients with variable impact on inpatient rehabilitation during the post-acute phase [7,8,12,13]. Increased age, injury intensity, subarachnoid hemorrhage, intraventricular hemorrhage, previous decompressive craniectomy and persistent low level of consciousness were identified as possible risk factors [9,13,14,15,16].

Hydrocephalus among SABI patients remains overlooked during inpatient rehabilitation [17,18,19].

The main problem with these patients is determining whether a ventriculomegaly is secondary to an atrophic process (hydrocephalus ex-vacuo) or due to an effective “active” hydrocephalus [3,7,20,21,22].

Discriminating whether a patient would benefit from a ventriculoperitoneal shunt (VPS) placement before the procedure is still difficult for a neurosurgeon and, subsequently, for the entire multidisciplinary team [7,21].

A careful albeit arduous selection of patients with ventriculomegaly as candidates for CSF diversion, is currently necessity for a multidisciplinary team in charge of these patients [1,2,8,21,23]. This is in fact a group of extremely fragile patients, characterized by a deteriorated neurological picture, often unable to report their symptoms, and suffering from numerous comorbidities. The problem is, therefore, not only to ascertain the diagnosis, but also to understand the actual usefulness of a possible intervention, taking the risk–benefit ratio into account.

In this paper, we analyzed the factors associated with post-surgical complications, length of stay, functional disability and cognitive impairment in SABI hydrocephalic patients treated with ventriculoperitoneal shunt placement before or during inpatient rehabilitation.

## 2. Materials and Methods

### 2.1. Study Design and Participants

This retrospective cohort study was performed in a tertiary referral specialized rehabilitation hospital in Italy. The study was approved by the local ethics committee (746-2020-OSS-AUSLIM—20123, 23/07/2020) and did not receive funding. Informed consent was obtained when possible or waived in accordance with the General Authorization of the Privacy Guarantor No. 09/2016 on observational retrospective studies.

Patients in the study were continuously admitted to the rehabilitation hospital with an intensive care unit and an early rehabilitation unit, from 1 January 2008 to 31 March 2017. Inclusion criteria were SABI of any etiological origin, diagnosed with hydrocephalus and treated with a shunt in the acute phase (before the admission) or during inpatient rehabilitation, age ≥ 18. VPS surgical procedure during inpatient rehabilitation was carried out at the Neurosurgery Department, Institute of Neurological Sciences of Bologna, IRCCS, Bellaria Hospital in Bologna. In the 9 years between 2008 and 2017, the neurosurgeons always remained the same and did not change their technology.

The suspicion of hydrocephalus emerged in patients who showed a progressive slowdown/reached plateau in neurological progress or a worsening of cognitive-behavioral functions in the presence of radiological signs of hydrocephalus (enlarged lateral and third ventricles, flattening or effacement of the cortical sulci) present on CT or MRI.

The presence of hydrocephalus was ascertained using radiological and clinical methods, including Evan’s Index, transependymal absorption with normal or reduced cerebral convexity spaces and small sylvian fissures on CT scans, and degree and extension of CSF flow void on T2-weighted magnetic resonance imaging scans.

In the last patients of the cohort, neurological improvement was observed after evacuative lumbar puncture.

Each patient was evaluated by a multidisciplinary team, consisting of one rehabilitation physician, one neurosurgeon, one speech therapist and one physiotherapist.

No exclusion criterion was applied.

Demographic characteristics, hydrocephalus etiology (traumatic, non-traumatic hemorrhagic and other, meaning anoxic, toxic-metabolic, or infectious), time between injury and hospitalization (time to hospitalization), time between injury and shunt placement (time to surgery), length of hospital stay, and destination after discharge (home, long-term care facility, another hospital, deceased) were recorded. The other variables related to VPS placement were pressure valve type used (fixed or programmable), incidence of clinical complications after surgery, cranioplasty and any related complications.

Moreover, in the subgroup of patients treated with VPS during inpatient rehabilitation, we distinguished time between admission and hydrocephalus diagnosis (time to diagnosis), and time between diagnosis and shunt placement (time between diagnosis and surgery).

### 2.2. Outcome Measures

The cognitive and behavioral assessment of patients was performed at admission and discharge using the Level of Cognitive Functioning Scale [24] (LCF), while the functional assessment was rated using the Disability Rating Scale (DRS) [25].

The LCF scale was developed as a useful and valid tool for the assessment of cognitive functioning in patients within the earliest phases of the post-coma state. The LCF was intended to provide a way of systematically describing and categorizing a patient’s present level of consciousness and cognitive and behavioral functioning. The LCF scale is one of the earlier developed scales used to assess cognitive functioning in post-coma patients. It was designed for use in the planning of treatment, tracking of recovery, and classifying of outcome levels. The scale generates a classification of patients in eight levels from 1 (non-responders) to 8 (purposeful–appropriate person); the higher the value, the better the cognitive improvement. The LCF scale was validated in Italian by Galeoto et al. [26].

The DRS is a 30-point scale with 8 areas of functioning: eye-opening; verbalization; motor response; level of cognitive ability for daily activities of feeding, toileting, and grooming; overall level of dependence; and employability. Each area of functioning is rated on a scale from 0 to 3, 0 to 4, or 0 to 5, with a higher score representing a lower level of functioning. Scores on each item are summed to yield a total score between 0 and 30, with a higher score indicating greater disability [27].

Data that reflect these aspects of function can predict the patient’s length of stay in rehabilitation and the ability to return to employment. We considered patients to be improved if they presented at least a 6-point score decrease in DRS (based on normative data, William and Smith [28]) and a 1-point increase in LCF.

### 2.3. Statistical Analysis

Age and rehabilitation assessment scales (LCF and DRS) were summarized using mean and standard deviation (SD), while timing variables (time to hospitalization, time to diagnosis, time to surgery and time between diagnosis and surgery) were summarized using median and interquartile range (Q1–Q3); categorical and dichotomous variables were summarized as absolute and relative frequencies.

Logistic regression models were used to identify the predictors of clinical complications after VPS placement. Histogram plots and Shapiro–Wilk tests were used to assess normality in the distribution of continuous outcome measures (LOS, DRS and LCF scores). Linear regression models were used to investigate the demographic and clinical predictors of these outcomes. In each model, predictors associated with outcome at a significance level *p* < 0.10 in univariate analyses were included in a multiple regression analysis. A backward stepwise procedure was used to obtain a final model including only significant predictors. Moreover, Hochberg’s correction for multiple comparisons was applied.

Analyses were performed using Stata statistical software version 15 (Stata Corp, College Station, TX, USA). The level of statistical significance was set at 0.05.

## 3. Results

A total of 82 out of the 621 SABI patients admitted to the rehabilitation institute during the study period met the inclusion criteria. Patients were aged between 18 and 82 years (mean ± SD 49.7 ± 17.8 years), 49% male, with non-traumatic hemorrhagic (55%), traumatic (35%), or other (10%) etiology (Table 1). In 23 patients, shunt placement was performed before admission to inpatient rehabilitation, in 59 (72%) during hospitalization (Figure 1).

The clinical features of patients that led to the diagnosis of hydrocephalus are described in Appendix A.

Time between injury and admission was 87 (71–130) days in the group treated before admission and 36 (26–61) days in the group treated during rehabilitation (*p* < 0.001). Conversely, time between injury and shunt placement was lower in the group treated before than in those treated after admission (56 (35–86) vs. 151 (89–207) days, *p* < 0.001). In the subgroup treated during rehabilitation, the median time to hydrocephalus diagnosis was 40 (18–89) days and the median time between diagnosis and VPS placement was 32 (20–56) days (Table 1).

Complications after VPS placement occurred in 15 patients (18%), with no significant difference between the group treated before admission and the group treated during hospitalization (9% vs. 22%, *p* = 0.213). Factors associated with VPS complications in univariate logistic regression were male gender, fixed pressure valve type and increased time to hospitalization (Table 2).

In the multiple regression model, only age and pressure valve type remained significant. The risk of developing complications after shunt placement was higher among patients with fixed pressure valves compared with those with programmable pressure valves (OR = 16.1, 95%CI: 3.1–84.4, *p* = 0.002) and decreased with age (OR = 0.94, 95%CI: 0.90–0.99, *p* = 0.008) (Table 2).

Sixteen patients (20%) underwent cranioplasty and six of them experienced related complications, including subdural hematoma (*n* = 3), subdural hygroma (*n* = 1) and infections (*n* = 3). Two patients required VPS removal. Overall, 46 patients (56%) were discharged at home (Table 1).

### 3.1. Factors Associated with Length of Hospital Stay

In univariate analyses, younger age, traumatic etiology, a longer time between injury and VPS placement and a programmable pressure valve type were associated with a longer LOS (Table 3).

In multiple regression, LOS increased with the increase in time to surgery (b = 0.38, 95% CI: 0.11–0.64, *p* = 0.006) and decreased with age (b = −0.29, 95%CI: −0.46–−0.12, *p* = 0.002), while a programmable pressure valve type was no longer significant (Table 3).

### 3.2. Rehabilitation Outcomes and Associated Factors

A significant improvement in cognitive functioning, from 3.0 ± 1.3 to 4.8 ± 1.7 (mean difference, 1.9 ± 1.3, *p* < 0.001), and a reduction in disability scores, from 21.1 ± 4.9 to 14.5 ± 6.0 (mean difference, 6.7 ± 4.3, *p* < 0.001) were found between admission and discharge. Overall, 56.3% achieved functional improvement at discharge and 88.7% improved in cognitive function. No patient worsened and only two patients (2.5%) showed no change in disability (DRS) and 9 (11.3%) in cognitive functioning (LCF) scores.

In univariate analyses, better cognitive functioning at discharge was associated with better functioning on admission, higher age, shorter LOS and the absence of cranioplasty (Table 4); in multiple regression analysis, only score on admission and the absence of cranioplasty remained significant (Table 4).

In univariate analyses, higher disability was associated with a higher disability on admission, lower age and longer LOS while, in multiple regression analysis, the scores on admission, LOS and cranioplasty were significant (Table 5).

### 3.3. Subgroup Analysis on Patients Undergoing VPS during Rehabilitation

In the subgroup of patients treated with VPS during rehabilitation (*n* = 59), time to diagnosis and time between diagnosis and surgery were unrelated to VPS complications (Appendix A).

## 4. Discussion

Hydrocephalus has been recognized as a frequent complication affecting the post-acute phase of inpatient rehabilitation after SABI [1,2,3,12,29]. In our study, we found an incidence of 13.2% (82 out of 621 patients admitted). The general condition of such patients could frequently impede a correct and timely diagnosis: patients cannot often report their symptoms and it might not be easy detecting signs and symptoms when dealing with SABI patients [1,2,5,12,30].

A persistent disorder of consciousness combined with a decline in neurobehavioral functions, or a plateaued clinical recovery discordant with injury severity as well as clinical improvement in behavioral or motor capacities after an evacuative lumbar puncture, should suggest a diagnosis of hydrocephalus. Radiological findings should also be congruous [3,5,13,30].

The reliability of the above-mentioned clinical features was supported by a clear improvement after the lumbar puncture, as adopted for the latter patients.

Our data showed that most of the patients shunted for hydrocephalus improved in their cognitive/behavioral (88.7%) and functional state (56.3%) through intensive and comprehensive rehabilitation programs and it was possible to discharge more than one half of them to their homes.

We found, on average, a decrease from a disability score of 21.2 upon admission to a score of 14.5 at discharge, which means the transition from vegetative state to severe disability with high dependence and low social reintegration. To the best of the authors’ knowledge, few previously published studies used LCF and DRS as outcome measures in SABI patients with hydrocephalus. Weintraub et al. [5], in a study performed in a rehabilitation hospital on 59 traumatic brain injured patients with PTH, used the Rancho Los Amigos Scale, also known as LCF, as one of the outcome measures. At discharge, the median RLAS score was 6 (range 3–7), with a median 2 point (range 0–4) increase from admission. A longer time from injury to VPS placement was unrelated with RLAS score at discharge or with RLAS change during rehabilitation.

Mazzini et al. [6] followed 140 PTH and found positive effects of rehabilitation on behavioral and functional outcomes measured by the Glasgow Outcome Scale, DRS, FIM^™^ and Neurobehavioral Rating Scale at 1-year postinjury follow-up.

Kammersgaard et al. [29] found that three-quarters of PTH emerged during rehabilitation and the condition of vegetative state upon referral rehabilitation raised the relative risk for PTH more than two-fold. The authors used RLAS to measure the level of consciousness and they considered the condition of vegetative state when the score was on level 2.

However, in SABI patients, LCF and DRS are commonly used in clinical practice as well as in research [26,31,32].

Many studies in the literature support the effectiveness of shunt for hydrocephalus following SABI with success percentages encouraging its treatment for these truly complex patients [5,11,13,17,23,33,34,35], but timing of treatment remains controversial [6,7,13,34].

At least for PTH patients, the literature is inconsistent regarding the impact of time from event to shunt on the outcome. Kowalski et al. [13] found that earlier shunting was associated with better outcome during rehabilitation. Conversely, Kim et al. [36] reported no significant relation concerning this interval time in their patient series and Sheffler [37] presented a case report of a PTH patient that improved 11 months after the trauma. SABI patients had a severe brain tissue damage that is expected to produce CSF circulation and absorption disturbance [6,38]. The different outcome after shunt depends on the severity of the primary brain injury after CSF circulation has been treated by shunt implantation. The recovery of patient and restoration of CSF circulation is a long-term process, especially for those with a more severe primary injury. The severity of hydrocephalus, as suggested by Wang et al. [18] in a study reporting a predictive model of outcome in PTH with severe conscious disturbance, can reflect the status of CSF circulation.

Our findings indicate that a longer time from event to shunt is associated with a longer LOS and, therefore, with a more complex rehabilitation program.

While time to diagnosis depends on the ability to intercept the early clinical manifestations of hydrocephalus, time to surgery depends on logistics and organization. At least for patients who underwent VPS placement during inpatient rehabilitation, surgical procedures were carried out at Bellaria Hospital, Institute of Neurological Sciences of Bologna, IRCCS. In these patients, we found an increase in LOS because of an increase in time between hydrocephalus diagnosis and treatment. A better organization, with prioritization in the surgical waiting list, could reduce the time from diagnosis to treatment.

LOS was longer for younger people. If LOS represents the recovery time, it could appear that younger people need more time to achieve the same recovery than older patients. As a matter of fact, elderly people tend to receive less therapy. “Discrimination” against the elderly population has its source in various issues: older people will live a shorter time with whatever skills taught to them during rehabilitation. They will not have a need for advanced skills because they are retired or likely to retire soon. They have also less cognitive reserve and, therefore, can only handle simpler and shorter therapy sessions [39]. On the other hand, a rehabilitation program for younger patients can have more comprehensive goals.

In the study carried out by Wang et al. [18], age < 50 years was an independent predictor for a good outcome, with younger patients showing a higher restoration of CSF circulation. On the contrary, in older people, the meningeal fibrosis seems more severe and should impact on CSF circulation and the ability of CSF absorption [40,41]. However, Tribl and Oder found that age at time of injury did not affect the outcome [16].

Our findings indicate that patients with poorer functioning at discharge had a longer LOS, after adjusting for scores on admission and cranioplasty. This could be due to the occurrence of other medical complications during rehabilitation. According to White et al. [4], hypertonia, agitation/aggression, urinary tract infection, and sleep disturbance are the most frequently reported problems in patients with traumatic disorders of consciousness. Pneumonia, gastrointestinal problems, and paroxysmal sympathetic hyperactivity are most likely to be severe. These complications were also common in our hospital and especially in patients with disorders of consciousness, who had poorer functioning and lower cognitive scores on admission. Active medical management takes time, prolonging length of stay and, unfortunately, does not affect the functional outcome.

Programmable pressure valves for shunt surgery have met surgeons’ enthusiasm since their introduction [42,43]. In our study, the risk of developing complications after shunt placement was 16 times higher among patients with fixed pressure valves compared with those with programmable pressure valves, differently from other studies [15,44].

The large cranial defect is one of the main indications for cranioplasty that have both cosmetic and therapeutic implications [2]. Surgical decompression itself can increase the incidence of hydrocephalus. For these patients, a VPS is usually necessary before cranioplasty. However, after placement of VPS, the patients may develop severe sinking of the scalp flap over the skull defect [14]. Consequently, when performing a cranioplasty, such depressed defects would increase the difficulty of the cranioplasty and the possibility of surgical complications (hematomas and intracranial infections). This problem was overcome in such patients by temporarily decreasing the shunt outflow by adjusting the shunt pressure using a programmable pressure valve to allow the expansion of a depressed scalp flap and facilitate the subsequent cranioplasty [45].

We found 11 patients who underwent cranioplasty, of whom five presented complications.

Among shunt complications, nine out of 15 patients were serious (infection and need for surgical revision). However, we did not find a significant relationship between complications and the other outcomes (LOS and rehabilitation scores). We found, conversely, a significant association between cranioplasty and poorer rehabilitation outcomes (motor and cognitive) at discharge, after adjusting for rehabilitation scores on admission and LOS.

### Limitations

The limitations of the present study include its retrospective nature and the heterogeneity of patients.

Data relating to the causes of hydrocephalus other than post-traumatic or post-hemorrhagic were missing because of record incompleteness.

Another limitation of the study was the absence of indication about the severity of the hydrocephalus to determine if this could be a confounding factor influencing the outcome after shunt.

## 5. Conclusions

Hydrocephalus represents one of the main complications among Severe Acquired Brain Injury patients and could variably affect their post-acute phase of rehabilitation. Our retrospective study showed a significant improvement in cognitive/behavioral and functional outcomes in patients who underwent ventriculoperitoneal shunt placement before or during inpatient rehabilitation. We found a significant relationship between time to shunt and length of stay. Moreover, poorer outcomes at discharge were associated with higher length of stay, likely due to the complexity of Severe Acquired Brain Injury patients.

The need of another elective procedure such as cranioplasty was associated with poorer outcomes. Moreover, use of fixed pressure valves in these patients was associated with a higher risk of developing complications in the overall sample.

Severe Acquired Brain Injury patients represent a group of fragile patients who require complex treatment from a well-trained multidisciplinary team. There is a need for larger prospective studies to standardize the diagnostic assessment protocol, the timing of treatment of hydrocephalus in Severe Acquired Brain Injury during inpatient rehabilitation and the value of reprogramming, in case of programmable pressure valve placement, on recovery at discharge.

## Figures and Tables

**Figure 1 brainsci-12-00003-f001:**
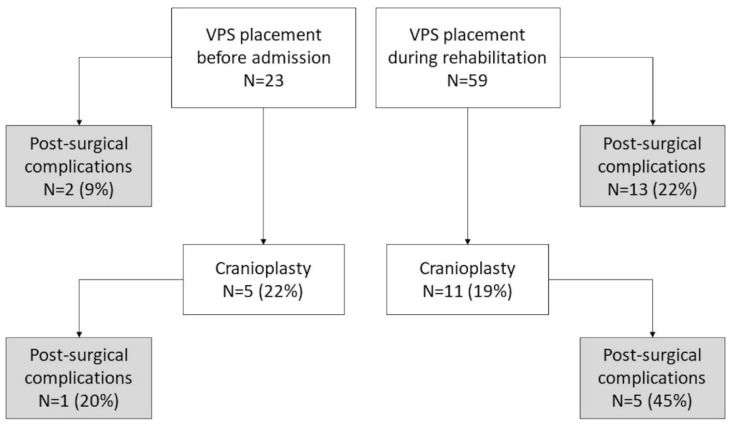
Study flowchart.

**Table 1 brainsci-12-00003-t001:** Characteristics of the study sample.

	Total Sample	VPS Placement before Admission	VPS Placement during Rehabilitation	*p*-Value
	*N* = 82	*N* = 23	*N* = 59	
Age, mean ± SD	49.7 ± 17.8	48.2 ± 18.9	50.3 ± 17.5	0.691
Male, *n* (%)	40 (49%)	10 (43%)	30 (51%)	0.549
Etiology, *n* (%)				0.475
Non-Traumatic Hemorrhagic	45 (55%)	14 (61%)	31 (53%)	
Traumatic	29 (35%)	6 (26%)	23 (39%)	
Other	8 (10%)	3 (13%)	5 (8%)	
Time to hospitalization (days) ^†^, median (IQ range)	48 (29–91)	87 (71–130)	36 (26–61)	**<0.001**
Time to surgery (days) ^§^, median (IQ range)	119 (64–186)	56 (35–86)	151 (89–207)	**<0.001**
Valve type, *n* (%)				0.852
Fixed	37 (45%)	10 (43%)	27 (46%)	
Programmable	45 (55%)	13 (57%)	32 (54%)	
VPS Complications, *n* (%)	15 (18%)	2 (9%)	13 (22%)	0.213
Cranioplasty, *n* (%)	16 (20%)	5 (22%)	11 (19%)	0.762
Cranioplasty Complications *	6 (38%)	1 (20%)	5 (45%)	
LCF, mean ±SD				
Upon admission	3.0 ± 1.3	3.3 ± 1.4	2.8 ± 1.3	0.178
At discharge	4.8 ± 1.7	5.0 ± 1.7	4.7 ± 1.8	0.478
DRS, mean ± SD				
Upon admission	21.2 ± 4.8	19.8 ± 4.7	21.7 ± 4.8	0.069
At discharge	14.5 ± 6.0	13.7 ± 5.8	14.8 ± 6.0	0.445
Length of hospital stay (days), median (IQ range)	221 (160–317)	189 (125–317)	227 (165–325)	0.297
Discharge, *n* (%)				0.263
At home	46 (56%)	17 (74%)	29 (49%)	
Hospital	20 (24%)	3 (13%)	17 (29%)	
Long-term Care Facility	14 (17%)	3 (13%)	11 (19%)	
Deceased	2 (2%)	0 (0%)	2 (3%)	

^†^ indicates time between injury and hospital admission; ^§^ indicates time between injury and VPS placement; * among patients undergoing cranioplasty surgery.

**Table 2 brainsci-12-00003-t002:** Factors associated with VPS complications: results from logistic univariate and multiple regression analyses. Significant associations are shown in boldface.

	Univariate Regressions		Multiple Regression	
	OR (95%CI)	*p*-Value	OR (95%CI)	*p*-Value
Age	0.97 (0.94–1.00)	0.099	0.94 (0.90–0.99)	**0.008**
Male	3.6 (1.0–12.5)	**0.043**		
Etiology		0.754		
Non-Traumatic Hemorrhagic (ref. cat.)	1.0			
Traumatic	1.4 (0.4–4.7)			
Other	1.8 (0.3–10.9)			
Time to hospitalization (weeks) ^†^	1.06 (1.00–1.13)	**0.039**		
Time to surgery (weeks) ^§^	1.02 (0.97–1.07)	0.450		
VPS during rehabilitation	3.0 (0.6–14.3)	0.176		
Fixed-type valve	6.7 (1.7–26.1)	**0.006**	16.1 (3.1–84.4)	**0.002**

^†^ indicates time between injury and hospital admission, and one patient with time >600 days was excluded; ^§^ indicates time between injury and VPS placement, and two patients with time >600 days were excluded.

**Table 3 brainsci-12-00003-t003:** Factors associated with LOS (weeks): results from linear univariate and multiple regression analyses. Significant associations are shown in boldface.

	Univariate Regressions		Multiple Regression	
	b (95%CI)	*p*-Value	b (95%CI)	*p*-Value
Age	−0.27 (−0.45–−0.09)	**0.004**	−0.29 (−0.46–−0.12)	**0.002**
Male	5.39 (−1.13–11.91)	0.104		
Etiology		**0.007**		
Non-Traumatic Hemorrhagic (ref. cat.)	1.00			
Traumatic	10.37 (3.59–17.15)			
Other	10.30 (−0.51–21.11)			
Time to hospitalization (weeks) ^†^	0.32 (−0.06–0.70)	0.101		
Time to surgery (weeks) ^§^	0.40 (0.12–0.68)	**0.006**	0.38 (0.11–0.64)	**0.006**
VPS during rehabilitation	2.96 (−4.36–10.29)	0.423		
Valve type fixed	−7.09 (−13.58–−0.61)	**0.032**		
VPS Complications	2.94 (−5.80–11.69)	0.505		
Cranioplasty	−0.93 (−9.46–7.61)	0.829		

Length of hospital stay is expressed in weeks, one patient with length of hospital stay >900 days was excluded; ^†^ indicates time between injury and hospital admission, one patient with time >600 days was excluded; ^§^ indicates time between injury and VPS placement, two patients with time >600 days were excluded.

**Table 4 brainsci-12-00003-t004:** Factors associated with cognitive function (LCF score) at discharge: results from univariate and multiple linear regression analyses. Significant associations are shown in boldface.

	Univariate Regressions		Multiple Regression	
	b (95%CI)	*p*-Value	b (95%CI)	*p*-Value
Age	0.02 (0.00–0.04)	**0.032**		
Male	−0.46 (−1.22–0.31)	0.239		
Etiology		0.164		
Non-Traumatic Hemorrhagic (ref. cat.)	1.00			
Traumatic	−0.75 (−1.56–0.07)			
Other	−0.73 (2.12–0.66)			
Time to hospitalization (weeks) ^†^	−0.01 (−0.06–0.04)	0.697		
Time to surgery (weeks) ^§^	−0.02 (−0.05–0.02)	0.340		
VPS during rehabilitation	−0.31 (−1.16–0.55)	0.478		
Valve type fixed	0.06 (−0.73–0.84)	0.885		
VPS Complications	−0.91 (−1.91–0.09)	0.073		
Cranioplasty	−1.03 (−1.97–−0.09)	**0.032**	−0.85 (−1.55–−0.14)	**0.019**
Length of hospital stay (weeks) *	−0.04 (−0.07–−0.02)	**0.001**		
Score on admission	0.90 (0.69–1.12)	**<0.001**	0.89 (0.68–1.10)	**<0.001**

Two patients who died before discharge were excluded; ^†^ indicates time between injury and hospital admission, one patient with time >600 days was excluded; ^§^ indicates time between injury and VPS placement, two patients with time >600 days were excluded; * one patient with length of hospital stay >900 days was excluded.

**Table 5 brainsci-12-00003-t005:** Factors associated with DRS score at discharge: results from univariate and multiple linear regression analyses. Significant associations are shown in boldface.

	Univariate Linear Regressions		Multiple Regression	
	b (95%CI)	*p*-Value	b (95%CI)	*p*-Value
Age	−0.08 (−0.15–−0.01)	**0.029**		
Male	1.50 (−1.16–4.16)	0.265		
Etiology		0.610		
Non-Traumatic Hemorrhagic (ref. cat.)	1.00			
Traumatic	1.41 (−1.45–4.28)			
Other	0.99 (−3.88–5.87)			
Time to hospitalization (weeks) ^†^	0.04 (−0.12–0.20)	0.636		
Time to surgery (weeks) ^§^	0.06 (−0.06–0.19)	0.301		
VPS during rehabilitation	1.14 (−1.81–4.09)	0.445		
Valve type fixed	−1.89 (−4.56–0.78)	0.162		
VPS Complications	2.12 (−1.37–5.62)	0.230		
Cranioplasty	3.02 (−0.26–6.30)	0.071	2.66 (0.33–4.99)	**0.026**
Length of hospital stay (weeks) *	0.15 (0.07–0.24)	**0.001**	0.08 (0.01–0.14)	**0.026**
Score on admission	0.87 (0.67–1.07)	**<0.001**	0.79 (0.59–0.99)	**<0.001**

Two patients who died before discharge were excluded; ^†^ indicates time between injury and hospital admission, one patient with time >600 days was excluded; ^§^ indicates time between injury and VPS placement, two patients with time >600 days were excluded; * one patient with length of hospital stay >900 days was excluded.

## Data Availability

Data are available from the first author upon reasonable request.

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
