# Peer review of "From Shunt to Recovery: A Multidisciplinary Approach to Hydrocephalus Treatment in Severe Acquired Brain Injury Rehabilitation"

_brainsci, 2021, doi:10.3390/brainsci12010003_

Round 1

Reviewer 1 Report

The authors attempt to examine the relationship between shunt surgery for hydrocephalus and functional outcomes in patients with severe acquired brain injury. However, the rationale and methodology of this study have been insufficiently justified. The quality of the manuscript is also not satisfactory.    1.Regarding the etiology of brain injury (Table 1), the authors list 3 categories: traumatic, hemorrhagic, and other. Does this imply that the traumatic cases were non-hemorrhagic?    2.The statistical methodology used in this study seems questionable, and it needs to be explained more clearly. For example, in some analysis (such as in Table 3) nonlinear regression model is used. However, the rationale for using nonlinear, rather than linear, model in that context is not explicitly stated, and the authors have not provided the details of their nonlinear regression model. These will make it difficult for the readers to understand and interpret the study results.    3.The discussion is poorly organized and often just reiterates the results (for example, page 12, line 361-366). Also, in the first paragraph on page 12, the authors mentioned about the different finding between this and other studies without further discussion; it would be better if the authors could elaborate more on this point.    4.(Last sentence of the abstract). It should be noted that craniectomy is not equal to cranioplasty, and it is the latter (i.e., cranioplasty) that is mainly examined in this study.    5.The manuscript is not well prepared. Just to name a few,  1) (page 1, line 16) The abbreviation "SABI" is not predefined. 2) (page 2, line 60) "patients may mimic NPH-like symptoms ....", it would be better stated as "patients may present with symptoms mimicking NPH ....", Also, NPH, presumably normal pressure hydrocephalus for short, is not predefined.  3) (page 2, line 67) Reference citations are not in bracket.

Author Response

TO REVIEWER 1

Comments and Suggestions for Authors

The authors attempt to examine the relationship between shunt surgery for hydrocephalus and functional outcomes in patients with severe acquired brain injury. However, the rationale and methodology of this study have been insufficiently justified. The quality of the manuscript is also not satisfactory.   

1.Regarding the etiology of brain injury (Table 1), the authors list 3 categories: traumatic, hemorrhagic, and other. Does this imply that the traumatic cases were non-hemorrhagic?   

Thank you for the review, your comment gave us the opportunity to improve the quality of our paper. We modified the tables in order to clear out this information: non traumatic hemorrhagic instead of hemorrhagic.

2.The statistical methodology used in this study seems questionable, and it needs to be explained more clearly. For example, in some analysis (such as in Table 3) nonlinear regression model is used. However, the rationale for using nonlinear, rather than linear, model in that context is not explicitly stated, and the authors have not provided the details of their nonlinear regression model. These will make it difficult for the readers to understand and interpret the study results. 

We thank the reviewer for noticing this point. We have now explicitly stated how we assessed normality in the frequency distribution of continuous variables in the methods section (lines 162-163). Because length of stay exhibited only a modest non-significant skewness, we have rerun the regression analyses comparing the results of non-linear regression models assuming that the dependent variable had a negative binomial distribution with those obtained assuming a normal distribution. We found that results were very similar, therefore we decided to apply linear regression analysis even for investigating factors associated to length of stay (lines 164-165). Results in table 3 were updated accordingly (table 3 and lines 228-230). 

3.The discussion is poorly organized and often just reiterates the results (for example, page 12, line 361-366). Also, in the first paragraph on page 12, the authors mentioned about the different finding between this and other studies without further discussion; it would be better if the authors could elaborate more on this point.   

According to the reviewer, we corrected the repetition in this way: Among shunt complications, 9/15 were serious (infection and need for surgical revision).

Thanks to the reviewer’s suggestion, we were able to add a paragraph in the discussion: These complications were also common in our hospital and especially in patients with disorders of consciousness, who had poorer functioning and lower cognitive scores on admission. An active medical management takes time, prolonging length of stay and, unfortunately, does not affect the functional outcome.

4.(Last sentence of the abstract). It should be noted that craniectomy is not equal to cranioplasty, and it is the latter (i.e., cranioplasty) that is mainly examined in this study.   

We thank the reviewer for this suggestion. We changed craniectomy with cranioplasty. The cranioplasty is the ideal consequence of surgical decompression as we indicated in the discussion, but according to the reviewer, we corrected the abstract because of the cranioplasty was mainly investigated in our study.

5.The manuscript is not well prepared. Just to name a few, 

1) (page 1, line 16) The abbreviation "SABI" is not predefined.

Thank you, we have now put into brackets right after Severe Acquired Brain Injury the acronym SABI at line 15.

2) (page 2, line 60) "patients may mimic NPH-like symptoms ....", it would be better stated as "patients may present with symptoms mimicking NPH ....", Also, NPH, presumably normal pressure hydrocephalus for short, is not predefined. 

Thank you for the revision that allows us to improve the quality of our paper. We modified according to your instructions: patients may present with symptoms mimicking Normal Pressure Hydrocephalus (NPH) at line 60.

3) (page 2, line 67) Reference citations are not in bracket.

Thank you, we have now put them in bracket.

Reviewer 2 Report

 Good study, suffering for a few methodological problems.

Retrospective analysis covered 9 years. Was a shunt technology used to manage hydrocephalus stable?

Use of acronyms in the conclusion statement should be avoided.

How the patients were selected for shunting. Radiology, gait analysis, any physiological measurements?

Reviewer 3 Report

The manuscript entitled “From shunt to recovery: a multidisciplinary approach to hydrocephalus treatment in severe acquired brain injury rehabilitation”. In the current manuscript, the authors have investigated the factors associated with post-surgical complications, length of stay, functional disability, and cognitive impairment in SABI (Severe Acquired Brain Injury) hydrocephalic patients treated with ventriculoperitoneal shunt placement before or during inpatient rehabilitation. They have suggested that Craniectomy, increased Length of stay (LOS), and fixed pressure valves are related to the worst outcomes. Also, the LOS was increased with increasing time to shunt and decreasing age.

The conclusion is comprehensive and highlights the strengths of the current study.

Best wishes.

Round 2

Reviewer 1 Report

The authors attempt to examine the relationships between shunt surgery for hydrocephalus and functional outcomes in patients with severe acquired brain injury. Some modifications were made to the initial manuscript in response to reviewers' comments. However, several issues remain inadequately addressed in the revised paper.   1.Has the severity of hydrocephalus been quantified or graded (such as using the ways described in the Methods, e.g., Evans' index)? The severity of hydrocephalus is a potential confounder when the relationships between shunt surgery and functional outcomes are examined.   2.Regarding the statistical methodology, several linear or logistic regression models (Table 2 to Table 5) were employed to assess the relationships between demographic and clinical variables and different outcome measures. I wonder if corrections for multiple testing should be performed to avoid type I errors.   3.The authors removed nonlinear regression models from the manuscript, but the abstract is not updated (line 21).    4.In response to reviewer's comment, the authors have replaced "craniectomy" in previous manuscript to "cranioplasty" in the current version. However, my opinion was that they should make it clear that the procedure done in each patient was craniectomy or cranioplasty, or both. Then analyze based on the real data.   5.Why did all the patients in this study receive VP shunt for hydrocephalus? Other CSF diversion methods, such as external ventricular drainage, could be more appropriate in some occasions. Had procedures other than VP shunt ever been done in these patients?   6.It is also difficult for me to understand the results of Table S2. The "time to diagnosis" and "time to surgery" should be continuous variables. How could the odds ratio be obtained?
